# Twin-Field Quantum Digital Signature with Fully Discrete Phase Randomization

**DOI:** 10.3390/e24060839

**Published:** 2022-06-18

**Authors:** Jiayao Wu, Chen He, Jiahui Xie, Xiaopeng Liu, Minghui Zhang

**Affiliations:** School of Information Science and Technology, Northwest University, Xi’an 710127, China; wujiayao@stumail.nwu.edu.cn (J.W.); chenhe@nwu.edu.cn (C.H.); 202032849@stumail.nwu.edu.cn (J.X.); liuxiaopeng@stumail.nwu.edu.cn (X.L.)

**Keywords:** quantum digital signature, twin-field key generation protocol, discrete-phase-randomized source

## Abstract

Quantum digital signatures (QDS) are able to verify the authenticity and integrity of a message in modern communication. However, the current QDS protocols are restricted by the fundamental rate-loss bound and the secure signature distance cannot be further improved. We propose a twin-field quantum digital signature (TF-QDS) protocol with fully discrete phase randomization and investigate its performance under the two-intensity decoy-state setting. For better performance, we optimize intensities of the signal state and the decoy state for each given distance. Numerical simulation results show that our TF-QDS with as few as six discrete random phases can give a higher signature rate and a longer secure transmission distance compared with current quantum digital signatures (QDSs), such as BB84-QDS and measurement-device-independent QDS (MDI-QDS). Moreover, we provide a clear comparison among some possible TF-QDSs constructed by different twin-field key generation protocols (TF-KGPs) and find that the proposed TF-QDS exhibits the best performance. Conclusively, the advantages of the proposed TF-QDS protocol in signature rate and secure transmission distance are mainly due to the single-photon interference applied in the measurement module and precise matching of discrete phases. Besides, our TF-QDS shows the feasibility of experimental implementation with current devices in practical QDS system.

## 1. Introduction

Digital signature is one of the kernel sciences behind classical cryptography [1]. It is particularly significant in modern communication as it can be used in a variety of applications, such as electronic mail, software distribution and financial transactions. The security of classical digital signature is guaranteed by computational difficulty assumption, which, however, will no longer be secure with the rapid development of quantum algorithms [2,3,4]. A full-fledged treatment for this issue towards quantum digital signature (QDS) [5] that paves a way to realize signature with information theory security is presented.

The first quantum signature protocol was introduced in 2001 [6], which can be considered as the original form of QDS. In general, earlier signature protocols [7,8,9,10,11,12] may impose several restrictions on QDS, such as non-destructive state comparison, long-time quantum memory and secure quantum channel, for obtaining a secure quantum signature. However, in practice, these requirements cannot be fully satisfied, resulting in security loopholes of real-life implementation. Subsequently, some practical QDS protocols [13,14,15,16,17,18] that do not attach these restrictions had been proposed and experimentally demonstrated.

For QDS protocols, a general scenario is that QDS can be divided into two assignments: distribution stage and messaging stage. The former uses the quantum part of quantum key distribution (QKD), i.e., key generation protocol (KGP), to distribute keys for users without further classical post-processing. The latter allows two receivers to verify the authenticity of a signature declaration. During the distribution stage, a KGP, such as BB84-KGP [19,20] or measurement-device-independent KGP (MDI-KGP) [21,22,23,24], will be adopted to generate correlated keys between users. However, the performance of both KGPs is restricted by the fundamental rate-loss limit (referred to as PLOB bound) [25,26], which is equal to $-\log_{2}\left(1-\eta \right)$, where η is the channel transmittance; this implies that the key generation rate can only vary with the channel transmittance linearly, asymptotically as 1.44η bits per channel use. Some newly proposed QKD protocols with better performance have accordingly improved the performance of KGPs, since KGP is regarded as a part of QKD. Recently, twin-field QKD (TF-QKD) protocol [27] has been proven to be capable of overcoming the PLOB bound. The reason is that the single-photon interference utilized in TF-QKD enables the key rate scale with the square root of the channel transmittance. Subsequently, various variants of original TF-QKD were designed and implemented for overcoming the PLOB bound [28,29,30,31,32,33,34,35,36,37,38,39].

In general, TF-QKD systems with decoy-state method [38,39,40,41,42] require users to emit coherent states with a continuous-phase-randomized source (CPRS); however, this is difficult to achieve in practice and may open a security loop. Remarkably, this issue can be solved by discrete-phase-randomized source (DPRS) instead and a rigorous security proof was already presented by Cao et al. [43]. The recently proposed TF-QKDs with discrete-phase-randomized source [44,45] closed the gap between theory and practice, and can be implemented with current optical devices. In particular, Zhang et al. [44] proposed a TF-QKD variant with *M* phase slices both in the code mode and the test mode. Currás-Lorenzo et al. [45] put forward a discrete-phase-randomized TF-QKD protocol with only two phases in the code mode, which provides a higher key rate than that in [44] since its key rate is not restricted by the sifting factor. Inspired by this work, we propose a practical discrete-phase-randomized TF-QDS protocol. In fact, TF-QDS is a kind of MDI-QDS performed at the single-photon level. We use a numerical approach to derive the bounds on parameters in the asymptotic case. For each given distance, we optimize the key rate over the signal intensity and decoy intensity of coherent pulses, and fix the vacuum intensity. For comparison, we plot the simulation results of various QDSs under the same experimental parameters and find that our TF-QDS can achieve higher signature rate and longer transmission distance compared with BB84-QDS [13] and MDI-QDS [16], when the number of discrete phase slices M≥6. The TF-QDS improves on current QDSs by overcoming the PLOB bound.

In addition, we compare the performance of our TF-QDS with two possible TF-QDSs constructed by two different KGPs: KGP with CPRS [38] and KGP with DPRS [44]. These two newly constructed protocols are called TF-QDS with CPRS and TF-QDS with DPRS, respectively. The simulation results demonstrate that our TF-QDS with M=6 can exceed the performance of TF-QDS with CPRS due to the exact matching of phases. Its signature rate is 5–15 times that of TF-QDS with CPRS when the transmission distance ranges from 100 km to 300 km, and the maximum signature distance obtainable can be increased by 5%. Furthermore, we compare the performance between our TF-QDS and TF-QDS with DPRS for four different *M*. The results show that, for the same *M*, our method can achieve better performance in terms of both signature rate and secure distance. The reason for this is that in our protocol, only two phases are encoded in the code mode and the key generation rate is not restricted by *M*. On the other hand, the increase of *M* tightens the upper bound of Eve’s side information estimated in the test mode. Therefore, for our protocol, the signature rate increases with *M*. However, in TF-QDS with DPRS, *M* phases are encoded in the code mode, so its signature rate tends to zero with the increase of *M*, which is caused by the filtering factor.

In this work, we devote Section 2 to the description of our TF-QDS. We review the TF-KGP [45] in Section 3. In Section 4, the security analysis is carried out. We give numerical simulation results in Section 5, and summarize our work in Section 6.

## 2. TF-QDS Protocol with Fully Discrete Phase Randomization

We consider a simple structure for our TF-QDS protocol to sign a one-bit message: a signer, Alice, generates a signature declaration with TF-KGPs performed by Alice–Bob and Alice–Charlie in the key distribution stage, and then transmits it to one of two receivers (Bob and Charlie), say Bob. Bob first verifies the signature and then forwards it to Charlie who then further verifies its authenticity in the messaging stage. The detailed process for our TF-QDS is illustrated in Figure 1. It is therefore clear that at most one party is dishonest in such a tripartite setting. If there is more than one dishonest party, then the real protocol will actually fail. We describe our TF-QDS protocol as follows.

### 2.1. Distribution Stage

(1)For each message m=0 or 1, Alice and Bob use the discrete-phase-randomized TF-KGP [45] to generate keys of length $n_{k}$, Alice holds the key $K_{m}^{AB}$ and Bob holds the key $K_{m}^{B}$. Similarly, Alice and Charlie perform the discrete-phase-randomized TF-KGP [45] to generate keys $K_{m}^{AC}$ and $K_{m}^{C}$, respectively. The detailed steps for key distribution can be found in Section 3. Alice’s signature for *m* is $Sig_{m}=\left(K_{m}^{AB},K_{m}^{AC}\right)$.(2)For each *m*, Bob and Charlie perform the symmetrization of keys. That is, Bob (Charlie) randomly chooses half of the bits in his key $K_{m}^{B}$ ($K_{m}^{C}$), called $K_{m,forward}^{B}$ ($K_{m,forward}^{C}$), and then sends these bits to Charlie (Bob) using an authenticated classical channel. The remaining bits in $K_{m}^{B}$ ($K_{m}^{C}$) are named $K_{m,keep}^{B}$ ($K_{m,keep}^{C}$). After the symmetrization of keys, Bob’s and Charlie’s keys are denoted as $S_{m}^{B}=\left(K_{m,keep}^{B},K_{m,forward}^{C}\right)$ and $S_{m}^{C}=\left(K_{m,keep}^{C},K_{m,forward}^{B}\right)$ respectively; therefore, Alice cannot distinguish whether a key is Bob’s or Charlie’s, which guarantees the security against repudiation. In addition, Bob (Charlie) can only obtain half of $K_{m}^{C}$ ($K_{m}^{B}$), which guarantees the security against forging.

### 2.2. Messaging Stage

(1)For signing a one-bit message *m*, Alice sends the signature declaration $\left(m,Sig_{m}\right)$ to Bob.(2)Bob checks and records the number of mismatches between the declaration $Sig_{m}$ and his key $S_{m}^{B}$. Particularly, Bob separately calculates the number of mismatches for the key $K_{m,keep}^{B}$ that received directly from Alice and the key $K_{m,forward}^{C}$ that forwarded by Charlie. If the number of mismatches for both parts is fewer than $S_{a}\left(n_{k}/2\right)$, he accepts the signature, where $S_{a}<1/2$ is a small threshold determined by experimental parameters and a desired security level.(3)Bob then forwards $\left(m,Sig_{m}\right)$ to Charlie.(4)Charlie checks the number of mismatches between $Sig_{m}$ and $S_{m}^{C}$. The verification method is similar to that performed by Bob, except with a different threshold $S_{v}$, where $0<S_{a}<S_{v}<1/2$. If the number of mismatches for $K_{m,keep}^{C}$ and $K_{m,forward}^{B}$ is fewer than $S_{v}\left(n_{k}/2\right)$, Charlie accepts $Sig_{m}$ as the original signature generated by Alice. It is worth noting that two different thresholds $S_{a}$ and $S_{v}$ are required to ensure the non-repudiation of QDS protocol.

## 3. TF-KGP

TF-KGP, as a part of QDS, is performed in pairs separately by Alice–Bob and Alice–Charlie to distribute keys without further error correction and privacy amplification. This section takes Alice and Bob as an example to review the TF-KGP with fully discrete phase randomization [45]. The distributed keys among Alice and Bob correspond to the keys $K_{m}^{AB}$ and $K_{m}^{B}$ described in Section 2.

### 3.1. Preparation

See the tasks below for the states of Alice’s and Bob’s delivering to Eve depending on their chosen mode for transmission. Alice and Bob choose the code mode and test mode randomly. The code mode is used for key generation and the test mode is used for parameter estimation. For the code mode, Alice (Bob) prepares a bit $k_{a}$ ($k_{b}$) randomly and generates a coherent state ${|\left(-1\right)}^{k_{a}}\sqrt{\mu }\rangle$ (${|\left(-1\right)}^{k_{b}}\sqrt{\mu }\rangle$), where μ is the signal intensity. For the test mode, Alice (Bob) prepares a discrete-phase-randomized coherent state $|\sqrt{\beta_{a}}e^{i\theta_{a}}\rangle$ ($|\sqrt{\beta_{b}}e^{i\theta_{b}}\rangle$), which is modeled by a random intensity $\beta_{a}$$\left(\beta_{b}\right)\in \left\{\beta_{0},\beta_{1},\mu \right\}$ ($\beta_{0}=0$ is a vacuum intensity and $\beta_{1}$ is a decoy intensity) and a random phase $\theta_{a}$$(\theta_{b})=2\pi m/M$ (m∈{0,1,2,⋯,M−1} and *M* is number of dividing the phase interval [0,2π) into slices [32]).

### 3.2. Measurement

An untrusted intermediate node, Eve, performs the single-photon interference on the incoming pulses through a 50:50 BS followed by two detectors SPD0 and SPD1. A successful round corresponds to only one detector being clicked, and is unsuccessful otherwise. Eve then announces the successful rounds publicly.

### 3.3. Sifting

Alice and Bob exchange their intensity and mode through an authenticated classical channel and retain data from those in which they have used the same mode. For rounds of code mode, Alice and Bob generate sifted keys $k_{A}$ and $k_{B}$. By disclosing *L* bits of sifted keys, they can calculate the quantum bit error rate (QBER), $E_{k}=\left(1/L\right)\sum k_{A}^{r}⊕k_{B}^{r}$, where $k_{A}^{r}$ and $k_{B}^{r}$ denote Alice’s and Bob’s exposed bits respectively, after which they are discarded. The remaining $n_{k}$ bits in $k_{A}$ and $k_{B}$ are Alice’s and Bob’s final keys $K_{m}^{AB}$ and $K_{m}^{B}$, respectively. It is necessary for Bob to flip his bits corresponding to rounds with SPD1 clicked. For rounds of test mode, Alice and Bob calculate the gains $\left\{Q_{\beta }\right\}$ in which they both select the same intensity and the same phase $\theta_{a}=\theta_{b}$, and the gains $\left\{Q_{\beta }^{-}\right\}$ in which they select the same intensity and the opposite phase $\theta_{a}=\theta_{b}\pm \pi$.

### 3.4. Parameter Estimation

Alice and Bob use the gains $\left\{Q_{\beta }\right\}$ and $\left\{Q_{\beta }^{-}\right\}$ to separately estimate the phase error rates $e_{ph,same}$ and $e_{ph,diff}$ according to the numerical method (see Appendix A), where $e_{ph,same}$ ($e_{ph,diff}$) indicates the phase error rate of successful code mode rounds in which Alice and Bob use the same (opposite) phase.

## 4. Security Analysis

We followed the method in Ref. [46] to estimate Eve’s smooth min-entropy [47] on Bob’s reserved key $K_{m,keep}^{B}$, and then use it to bound the probability that Eve makes errors less than a certain value.

Eve can obtain some information from parameter estimation and mode declaration. Here, we define κ and ζ as the classical information disclosed during parameter estimation and mode declaration, respectively. $K_{m,forward}^{B}$ is the extra information leaked to Eve under the case that Charlie is Eve. All these information is defined on one quantum system living in the Hilbert space, which is a combination of the following elements: κ, ζ and $K_{m,forward}^{B}$, as well as Eve’s ancilla quantum system following her general attack. The information that Eve obtains about $K_{m,keep}^{B}$ is summarized as *E*. Then, Eve’s smooth min-entropy with access to *E* is given by
(1)$$H_{min}^{\epsilon_{k}}\left(K_{m,keep}^{B}|E\right)\geq \frac{n_{k}}{2}\left(1-I_{AE}\right),$$
where the inequality holds up to $\log_{2}\left(1/\epsilon_{k}\right)$. Here, $I_{AE}$ denotes the information leaked to Eve, which can be bounded by the phase error rates $e_{ph,same}$ and $e_{ph,diff}$ as
(2)$$I_{AE}\leq \frac{1}{2}h\left(e_{ph,same}\right)+\frac{1}{2}h\left(e_{ph,diff}\right),$$
where h(x) is a Shannon binary entropy function $h\left(x\right)=-x\log_{2}x-\left(1-x\right)\log_{2}\left(1-x\right)$. The phase error rates cannot be directly observed from experiments, their estimation can be found in Appendix A. With the upper bound on $I_{AE}$, we can further evaluate Eve’s smooth min-entropy.

Secondly, according to the proposition in Ref. [16], the upper bound on the average probability that Eve’s eavesdropping makes at most *r* errors with the given smooth min-entropy is
(3)p≤∑t=0rnk2t2−Hminϵk(Km,keepB|E)+ϵk.
Furthermore, for large $n_{k}$, the probability for Eve to make fewer than *r* errors for any g>0 is given by
(4)P(Evemakesfewerthanrerrors):=p≤g,
except with probability at most
(5)$$\epsilon_{F}:=\frac{1}{g}\left(2^{-\frac{n_{k}}{2}\left[\left(1-I_{AE}\right)-h\left(2r/n_{k}\right)\right]}+\epsilon_{k}\right).$$
Thus, we can determine the condition that Eve can make fewer than *r* errors with a non-negligible probability as
(6)$$\left(1-I_{AE}\right)-h\left(2r/n_{k}\right)>0.$$
If this condition is met, the probability of Eve making fewer than *r* errors will be arbitrarily small by increasing the length of signature. We define $p_{E}$ as:(7)$$\left(1-I_{AE}\right)-h\left(p_{E}\right)=0.$$
A physical interpretation behind Equation (Equation 7) is that $p_{E}$ is the minimum error rate that Eve makes when guessing Bob’s key except with negligible probability $\epsilon_{F}$. The upper bound on QBER between Alice’s and Bob’s keys is $\bar{E_{k}}$. Therefore, as long as the condition of $p_{E}>\bar{E_{k}}$ is satisfied, we can obtain a secure signature by increasing $n_{k}$, which means that
(8)$$\left(1-I_{AE}\right)-h\left(\bar{E_{k}}\right)>0.$$

For demonstrating the security of TF-QDS protocol, we aim to show that the following three inherent properties for signature systems can be guaranteed [48].

### 4.1. Robustness

In the messaging phase, Bob would reject Alice’s signature declaration when the mismatch rate between $n_{k}/2$ bits received from either Alice or Charlie and Alice’s declaration is higher than $S_{a}$. The QBER (mismatch rate) $E_{k}$ between Alice’s and Bob’s keys can be estimated by utilizing *L* bits. According to the Serfling inequality [49], we can obtain the upper bound on QBER as follows:(9)$$\bar{E_{k}}\geq E_{k}+\tau \left(\frac{n_{k}}{2},L,\epsilon_{P}\right),$$
where
(10)$$\tau \left(\frac{n_{k}}{2},L,\epsilon_{P}\right)=\sqrt{\frac{\left(\frac{n_{k}}{2}-L+1\right)\ln\left(\frac{1}{\epsilon_{P}}\right)}{Ln_{k}}}.$$
This suggests that the upper bound on QBER is true except with a small probability $\epsilon_{P}$. The failure probability decays exponentially in the parameter *L* for any fixed value of the function τ. We set $\bar{E_{k}}:=\max\left\{\bar{E_{k,B}},\bar{E_{k,C}}\right\}$, where $\bar{E_{k,B}}$ and $\bar{E_{k,C}}$ correspond to the upper bounds on QBERs for Alice–Bob and Alice–Charlie, respectively. Here, we should make $S_{a}$ greater than $\bar{E_{k}}$, except with probability of at most $\epsilon_{P}$, so the probability of an honest abort is restricted to
(11)$$P\left(\mathrm{honest}\;\mathrm{abort}\right)\leq 2\epsilon_{p}.$$

### 4.2. Non-Repudiation

Non-repudiation means that the signature declaration generated by an original signatory is accepted by one receiver but rejected by the other.

For repudiation, the number of mismatches between Alice’s declaration $Sig_{m}$ and Bob’s key $S_{m}^{B}$ must be less than $S_{a}\left(n_{k}/2\right)$, and that between $Sig_{m}$ and Charlie’s key $S_{m}^{C}$ must be greater than $S_{v}\left(n_{k}/2\right)$. This suggests that Alice should make Bob accept her signature and make Charlie reject her signature. That is, a necessary condition for successful repudiation is that the mismatch rate between $Sig_{m}$ and $S_{m}^{B}$ is not equal to that between $Sig_{m}$ and $S_{m}^{C}$. In this protocol, the symmetrization of keys performed between Bob and Charlie makes their respective keys contain an equal error rate, resulting in security against repudiation. According to the results in Ref. [13], the probability of Alice’s successful repudiation can be bounded as
(12)$$P\left(\mathrm{repudiation}\right)\leq 2\exp[-\frac{1}{4}{\left(S_{a}-S_{v}\right)}^{2}n_{k}],$$
where
(13)$$S_{a}=\bar{E_{k}}+\frac{P_{E}-\bar{E_{k}}}{3},\;\;\;\;\;S_{v}=\bar{E_{k}}+\frac{2(P_{E}-\bar{E_{k}})}{3}.$$

### 4.3. Unforgeability

Forgery attack performed by a dishonest internal user is considered in our analysis since it is more convenient for insiders to perform a forgery attack than external attackers. Suppose that Bob wants to forge Alice’s signature: he needs to transmit Charlie a forged signature and make the number of mismatches contained in the forged signature fewer than $S_{v}\left(n_{k}/2\right)$. As mentioned above, $\bar{E_{k}}$ is the upper bound on QBER between Alice’s and Charlie’s keys, and $p_{E}$ indicates the minimum error rate that Bob makes errors associated with Charlie’s key. When Equation (Equation 8) holds, we choose $S_{v}$ such that $\bar{E_{k}}<S_{v}<p_{E}$. This suggests that there is a higher probability for Charlie to accept Alice’s original signature. On the contrary, Charlie will likely reject Bob’s forged signature, since the probability of Bob creating a forged signature with an error rate fewer than $S_{v}$ is restricted by Equation (Equation 4) as
(14)$$P(\mathrm{Bob}\;\mathrm{makes}\;\mathrm{fewer}\;\mathrm{than}\;S_{v}\left(n_{k}/2\right)\;\mathrm{errors}):=p\leq g,$$
except with probability at most $\epsilon_{F}$. If the estimation of parameters $E_{k}$ and $e_{ph}$ (containing $e_{ph,same}$ and $e_{ph,diff}$) fails, which separately occur with probabilities $\epsilon_{P}$ and $\epsilon_{ph}$, then we think for simplicity that Bob can successfully forge Alice’s signature. Thus, the probability of Bob’s successful forging can be bounded as
(15)$$P\left(\mathrm{forge}\right)\leq g+\epsilon_{F}+\epsilon_{P}+2\epsilon_{ph}.$$
This equation is valid for any choice of parameters greater than zero. The probability for Bob to forge a signature can be arbitrarily small by increasing $n_{k}$.

In summary, the security level is bounded as
(16)ξ=max{P(honestabort),P(repudiation),P(forge)}.

## 5. Numerical Simulation

In this section, we give simulation results for TF-QDS with two-intensity decoy-state method in the asymptotic scenario. The channel model given in Ref. [45] is employed to obtain observable gains and QBERs for TF-KGP. For simplicity, we assume that the channel is symmetrical for each pair of parties. For each given distance, we optimize the key rate over the signal intensity μ and the decoy intensity $\beta_{1}$ of coherent pulses, and fix the vacuum intensity $\beta_{0}=0$.

We plot the signature rate *R* as a function of the transmission distance with optimal values of intensities μ and $\beta_{1}$ for a given security level $10^{-8}$, as shown by dashed lines in Figure 2. The signature rate is defined as R=1/N, where *N* indicates the total number of pulses required to sign a 1-bit message given a certain security level. If we set a fixed security level, then the signature rate can be bounded by Equation (Equation 12) with parameters $e_{ph,same}\left(e_{ph,diff}\right)$, $\bar{E_{k}}$ and $P_{E}$. The optimal values of intensity μ with different numbers of phase slices *M* against the transmission distance can be found in Figure 3. These optimal values are obtained by maximizing the key rate of KGP. The experimental parameters used for numerical simulation refer to a recent experiment [50], which are listed in Table 1. We also give the simulation results of signature rate without performing any optimization for comparison. In this case, we plot the signature rate curves with fixed values of all intensities (μ=0.06, $\beta_{1}=10^{-4}$ and $\beta_{0}=0$), represented by solid lines in Figure 2. As depicted in Figure 2, numerical optimization yields a significant improvement in transmission distance compared to nonoptimization, especially when *M* is relatively small. In particular, the maximum signature distance reached with intensity optimization increased by more than two times when M=4.

Furthermore, Figure 2 shows the performance of TF-QDS for different *M*. The maximum signature distances when M=8 and M=12 have only slight differences under the same experimental parameters, corresponding to 317 km and 325 km, respectively. This suggests that it is not necessary to increase the number of phase slices to achieve significant improvements. On the other hand, for M=4, M=6 and M=8, there is a distinct improvement in terms of signature distance for larger *M*.

We perform a numerical simulation of M=8 to evaluate the potential impact of using more decoy states in terms of performance. Figure 4 shows the comparison of simulation results for TF-QDS with different numbers of decoy intensities. The results show that the signature rate and transmission distance of TF-QDS with two decoy states is close to that of three [51] (and four) decoy states. Therefore, for our TF-QDS protocol, the two-intensity decoy state is sufficient for practical usage, and there is no need to introduce more decoy states for longer transmission distances.

We compare the performance of our TF-QDS with BB84-QDS and MDI-QDS in Figure 5. For a fair comparison, we plot the signature rate curves of three QDSs by using the same experimental parameters without intensity optimization. The channel models given in Ref. [32] are utilized to calculate gains and QBERs for BB84-QDS and MDI-QDS. As shown in Figure 5, BB84-QDS shows the highest signature rate when the transmission distance is less than 45 km. Once the distance is more than 45 km, the signature rate of TF-QDS exceeds that of BB84-QDS. Compared with MDI-QDS, the signature rate of TF-QDS is always better than that of MDI-QDS when M≥6. In addition, TF-QDS can obtain a secure signature at longer transmission distance than BB84-QDS and MDI-QDS when M≥6. Among these QDS protocols, the maximum signature distance for TF-QDS is 520 km when M=12, whereas the maximum signature distances for BB84-QDS and MDI-QDS are 182 km and 334 km, respectively. This is because the measurement module of TF-QDS is realized with single-photon interference which requires only one photon to survive the loss of over half of the transmission distance. Twin-field approach overcomes the PLOB bound and significantly extends the secure signature distance.

Our TF-QDS protocol is built upon the TF-KGP in Ref. [45] where users emit coherent-states with a discrete-phase-randomized source in the test mode. We can further propose two possible TF-QDS protocols: TF-QDS with CPRS and TF-QDS with DPRS, which are separately constructed by KGP in Ref. [38] and KGP in Ref. [44], respectively. We compare the performance of our TF-QDS with two newly constructed TF-QDSs with the same experimental parameters given in Ref. [38].

Firstly, we compare the simulation results of our TF-QDS and TF-QDS with CPRS in Figure 6. Remarkably, the results show that our TF-QDS with only six discrete phase slices can exceed the performance of TF-QDS with CPRS. In detail, our protocol can achieve a secure signature at the maximum transmission distance of 408 km with M=6, while the maximum transmission distance for TF-QDS with CPRS is 388 km. Besides, its signature rate increases by more than one order of magnitude since 69 km. The reason for this improvement is that a tighter bound on the phase error rate can be obtained, since the phase post-selection in discrete version makes the users’ phases exactly matched.

Furthermore, we compare the performance between our TF-QDS and TF-QDS with DPRS for different *M*. Both TF-QDS protocols utilize a discrete-phase-randomized source, with the main difference being that, for the proposed TF-QDS, only two phases rather than *M* phases are encoded in the code mode. The simulation results are shown in Figure 7, where solid lines correspond to the results of our TF-QDS, and dashed lines correspond to the results of TF-QDS with DPRS. Figure 7 illustrates that our TF-QDS can deliver a higher signature rate than that of TF-QDS with DPRS for the same phase slice. The fact is that the signature rate of our protocol increases with *M*, while the signature rate of TF-QDS with DPRS approaches 0 as *M* increases, due to the sifting factor. Furthermore, our TF-QDS can transmit a longer signature distance when the same signature rate is obtained. The reason for this is that we can obtain a tighter bound on the phase error rate compared with TF-QDS with DPRS, as illustrated in Figure 8. Detailed data on signature rate *R* and transmission distance for both TF-QDSs are listed in Table 2.

## 6. Conclusions

We present a TF-QDS protocol with fully discrete phase randomization. Unlike most previous variants of QDS that emit weak coherent-state pulses with a continuous-phase-randomized source, our TF-QDS uses a discrete-phase-randomized source instead, which can be realized with common optical components and further applied in practical QDS systems. As well as this, the protocol had been proved to be secure against forging and repudiation.

For better performance, we optimize intensities of signal state and decoy state to improve the signature rate. We compare the performance of our TF-QDS with BB84-QDS and MDI-QDS by numerical simulation. The results demonstrate that our TF-QDS can achieve the best performance in terms of signature rate and secure transmission distance when the phase slices M≥6. Moreover, we provide a clear comparison between several possible TF-QDSs constructed by different TF-KGPs and find that our TF-QDS with M=6 already exceeds TF-QDS with CPRS due to the exact matching of phases. The signature rate is 5–15 times that of TF-QDS with CPRS when the transmission distance ranges from 100 km to 300 km, and its maximum signature distance obtainable increases by 5%. Furthermore, we compare the performance of our TF-QDS with TF-QDS with DPRS for four different *M*. The simulation results show that our TF-QDS achieves a higher signature rate and a longer secure distance than that of the TF-QDS with DPRS for the same *M*.

In summary, the proposed TF-QDS with fully discrete phase randomization is more feasible in experimental implementation; meanwhile, it is an effective solution for a higher signature rate over a longer transmission distance.

## Figures and Tables

**Figure 1 entropy-24-00839-f001:**
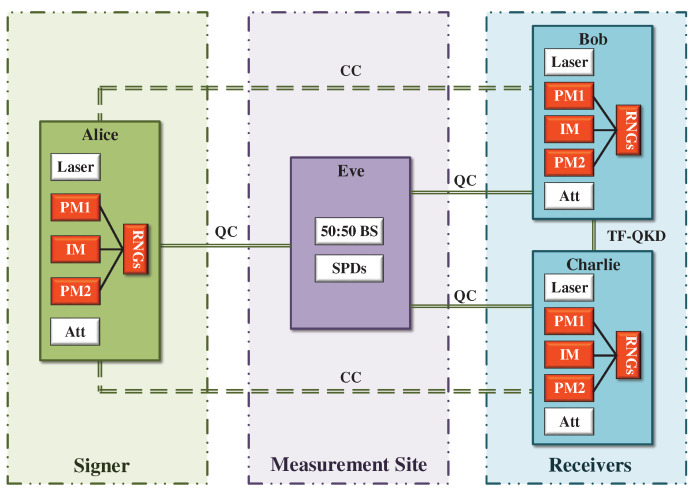
Schematic diagram of our TF-QDS. Alice and Bob (Alice and Charlie) prepare discrete-phase-randomized weak coherent-state pulses with a phase modulator (PM1). Intensity modulator (IM) is used to generate decoy states and PM2 is used to encode key bits. All the modulations are combined with random number generators (RNGs). The encoded pulses are attenuated by an attenuator (Att) and then sent out to the measurement site Eve. Alice’s and Bob’s (Alice’s and Charlie’s) pulses interfere at a 50:50 beam splitter (BS). The interference result is recorded with two single-photon detectors (SPDs). The solid lines represent the quantum channels (QCs) through which Alice and Bob (Alice and Charlie) use KGP to distribute keys. The dotted lines represent the authenticated classical channels (CCs) through which parties exchange and transmit some classical message. A TF-QKD link is shared between Bob and Charlie for performing the symmetrization of keys in full secrecy.

**Figure 2 entropy-24-00839-f002:**
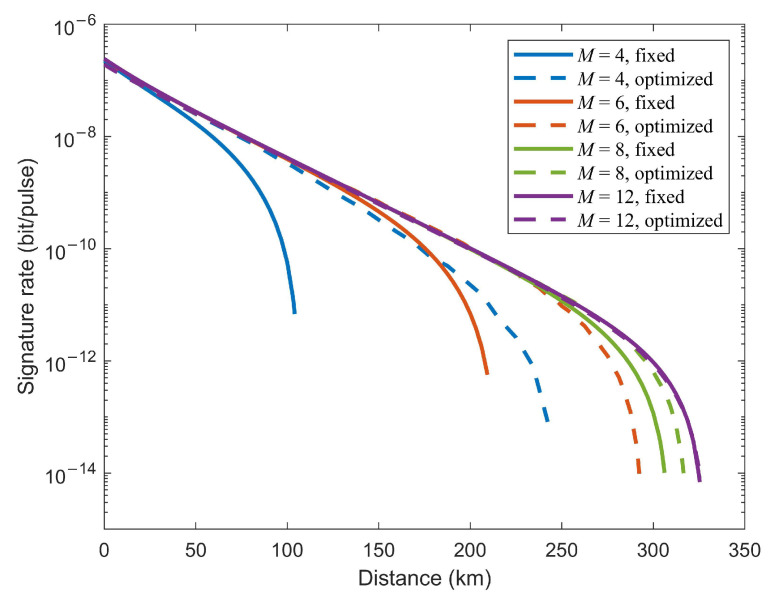
Signature rate vs. the transmission distance with optimal intensities (dashed lines) and fixed intensities (solid lines) for four different numbers of phase slices *M*. (M=4 blue, M=6 red, M=8 green, M=12 purple).

**Figure 3 entropy-24-00839-f003:**
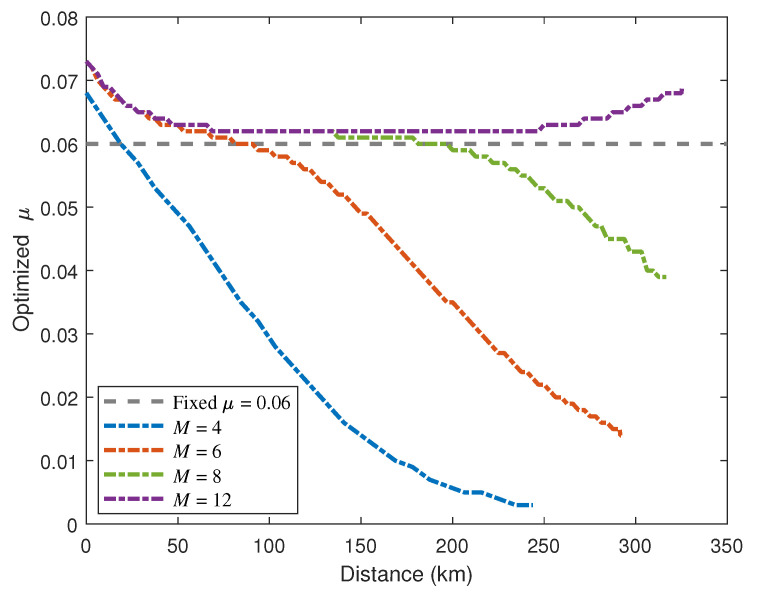
Optimal values of the signal intensity μ vs. the transmission distance for four different *M*.

**Figure 4 entropy-24-00839-f004:**
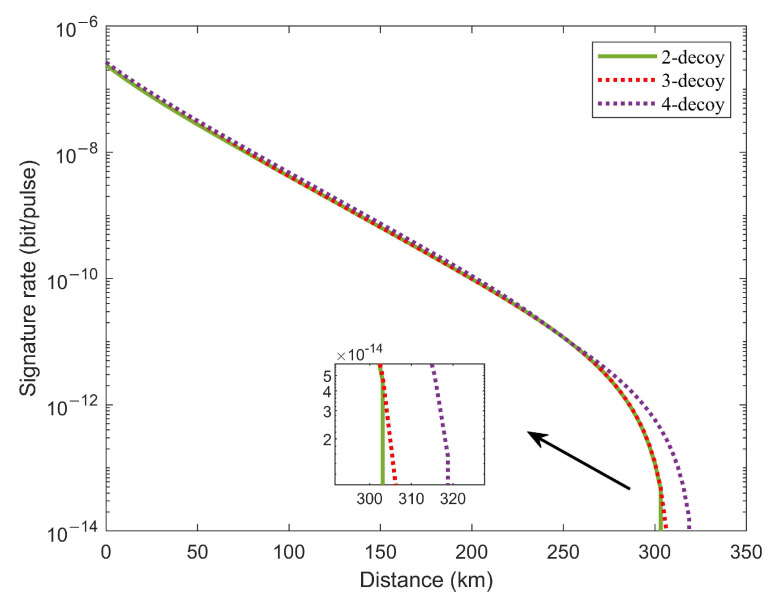
Signature rate versus the transmittance distance by using two-intensity (green solid line), three-intensity (red dotted line) and four-intensity (purple dotted line) decoy-state methods for M=8.

**Figure 5 entropy-24-00839-f005:**
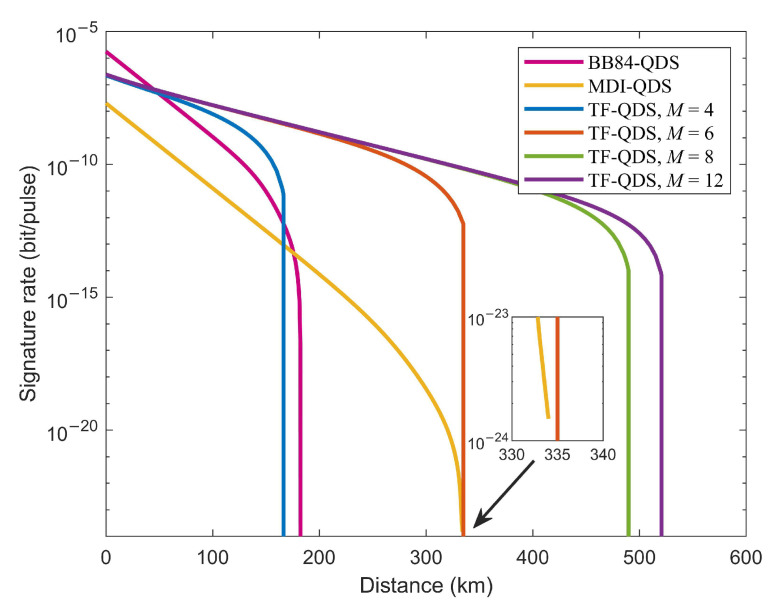
Results of our TF-QDS, BB84-QDS and MDI-QDS.

**Figure 6 entropy-24-00839-f006:**
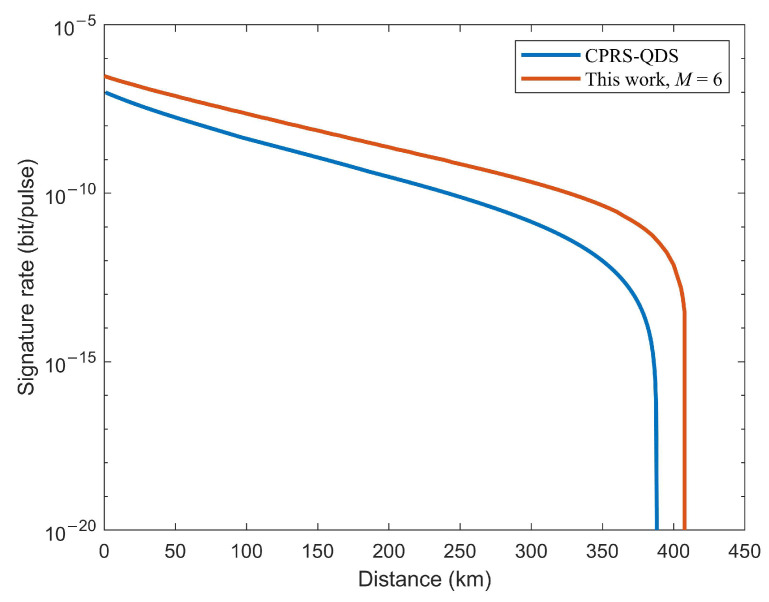
Results of our TF-QDS and TF-QDS with CPRS.

**Figure 7 entropy-24-00839-f007:**
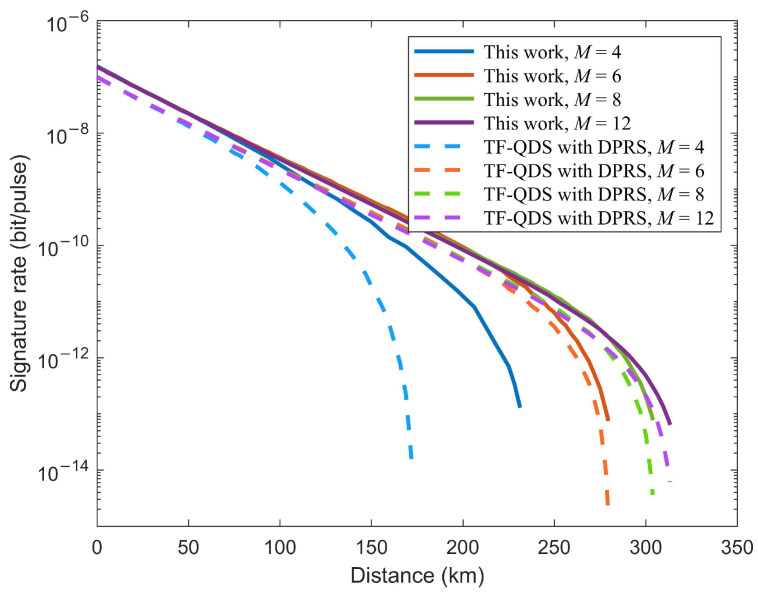
Results of our TF-QDS and TF-QDS with DPRS for four different *M*.

**Figure 8 entropy-24-00839-f008:**
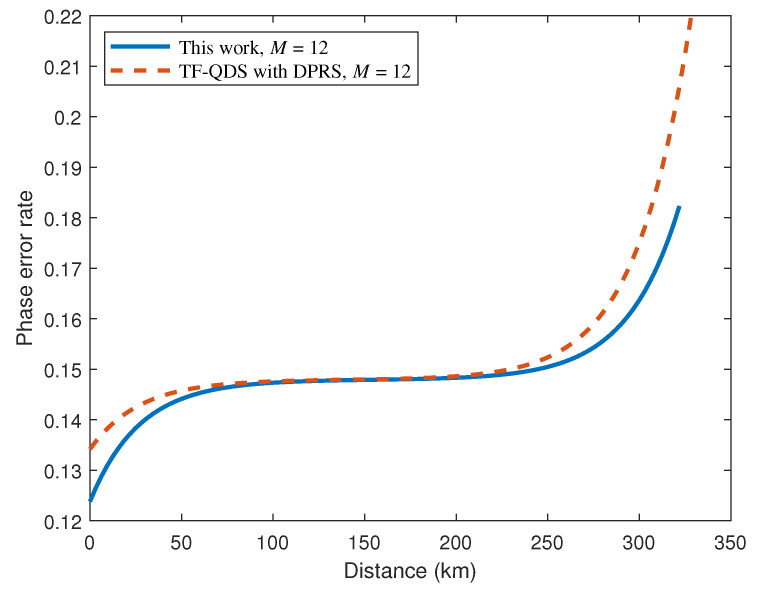
Comparison the upper bound of phase error rate between our TF-QDS and TF-QDS with DPRS for M=12.

**Table 1 entropy-24-00839-t001:** Parameter setting in simulation. α—loss coefficient of fiber at telecommunication wavelength; $p_{d}$—dark count rate of detectors; $\eta_{d}$—detection efficiency of detectors; $e_{d}$—optical misalignment error; *f*—error correction inefficiency.

α	$p_{d}$	$\eta_{d}$	$e_{d}$	*f*
0.16	$10^{-8}$	35%	2%	1.15

**Table 2 entropy-24-00839-t002:** Comparison of signature rate *R* and transmission distance for our work and TF-QDS with DPRS for different *M*. More general comparison results are shown in Figure 6. The second and third rows indicate the signature rates of two protocols at 100 km and 200 km, respectively. The fourth row shows the secure transmission distances when the signature rate is $10^{-12}$ and the bottom row gives the comparison of maximum transmission distances obtainable.

	Protocols	M=4	M=6	M=8	M=12
*R* at 100 km	This work	$3.37\times 10^{-9}$	$3.66\times 10^{-9}$	$3.50\times 10^{-9}$	$3.42\times 10^{-9}$
(bits/pulse)	TF-QDS with DPRS	$1.30\times 10^{-9}$	$2.36\times 10^{-9}$	$2.27\times 10^{-9}$	$2.23\times 10^{-9}$
*R* at 200 km	This work	$2.08\times 10^{-11}$	$9.31\times 10^{-11}$	$8.77\times 10^{-11}$	$8.30\times 10^{-11}$
(bits/pulse)	TF-QDS with DPRS	-	$5.75\times 10^{-11}$	$5.64\times 10^{-11}$	$5.38\times 10^{-11}$
Distance	This work	222.2	267.7	288.8	291.8
(km)	TF-QDS with DPRS	164.8	262.2	281.7	283.9
Maximum	This work	231.3	279.4	303.8	313.3
distance (km)	TF-QDS with DPRS	171.9	279.3	303.7	313.1

## Data Availability

Data are contained within the article.

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
