# Peer review of "Twin-Field Quantum Digital Signature with Fully Discrete Phase Randomization"

_entropy, 2022, doi:10.3390/e24060839_

Round 1

Reviewer 1 Report

LINE 25-27: the phrese is missing the verb,  I gues at the end of the lines  "is presented" must be added.

1) in paragraph 60 it is reported that the sifting factor does not affect the transmission rate ( Currás-Lorenzo et al [44] put forward a discrete-phase-randomized TF-QKD 61 protocol with only two phases in the code mode, which provides a higher key rate 62 than that in [43] since its key rate is not restricted by the sifting factor );  in paragraph 250 it is reported that  "The fact is that the signature rate of our protocol increases with M, while the signature rate of TF-QDS with DPRS approaches 0 as M increases, due to the sifting factor" ; I cannot reconcile the two apparently contradictory statements.
  2) in paragraph 168 where  log2(1/fflk ) is first introduced  , fflk is not defined; 3) I believe that the protocol is interesting and give advantages with respect to the proposed competing protocols, nonetheless i think that a more fair comparison would be with QKD protocols with higher dimensionality; so it would be interesting to have a comparison with qkd protocols with 6 or more states .

Author Response

Point 1: LINE 25-27: the phrase is missing the verb, I guess at the end of the lines "is presented" must be added.

Response 1: Thanks for your reminding, we had corrected this grammatical error.

Point 2: In paragraph 60 it is reported that the sifting factor does not affect the transmission rate (Currás-Lorenzo et al [44] put forward a discrete-phase-randomized TF-QKD 61 protocol with only two phases in the code mode, which provides a higher key rate 62 than that in [43] since its key rate is not restricted by the sifting factor );  in paragraph 250 it is reported that  "The fact is that the signature rate of our protocol increases with M, while the signature rate of TF-QDS with DPRS approaches 0 as M increases, due to the sifting factor" ; I cannot reconcile the two apparently contradictory statements.

Response 2: Considering the reviewer’s suggestion, we had added some descriptions to explain why our protocol is not restricted by the sifting factor. These statements “The reason is that in our protocol, only two phases are encoded in the code mode and the key generation rate is not restricted by M. On the other hand, the increase of M tightens the upper bound of Eve's side information estimated in the test mode. Therefore, for our protocol, the signature rate increases with M. However, in TF-QDS with DPRS, M phases are encoded in the code mode, so its signature rate tends to zero with the increase of M, which is caused by the filtering factor.” had been added on lines 83-88.

Point 3: in paragraph 168 where  log2(1/fflk ) is first introduced, fflk is not defined;

Response 3: This is a typographical error and we had corrected it on line 173.

Point 4: I believe that the protocol is interesting and give advantages with respect to the proposed competing protocols, nonetheless i think that a more fair comparison would be with QKD protocols with higher dimensionality; so it would be interesting to have a comparison with qkd protocols with 6 or more states .

Response 4: According to the comment of reviewer, we performed a fair comparison in Figure 4 between our TF-QDS protocol and the protocols using more decoy states, with the same experimental parameters. Some statements about the simulation results “We perform a numerical simulation of M=8 to evaluate the potential impact of using more decoy states in terms of performance. Figure 4 shows the comparison of simulation results for TF-QDS with different numbers of decoy intensities. The results show that the signature rate and transmission distance of TF-QDS with two decoy states is close to that of three[51] (and four) decoy states. Therefore, for our TF-QDS protocol, the two-intensity decoy state is sufficient for practical usage, and there is no need to introduce more decoy states for longer transmission distances.”were added on lines 219-225.

Reviewer 2 Report

I have read this interesting manuscript which contains publishable contents. The idea put forward of a TF-based QDS scheme can have experimental implications.

The authors study the protocol assuming the use of two-intensity decoy states.

I would like to see the potential impact of using more decoys in terms of performance.

Can we have an observable improvement by using a 3-intensity decoy state method?

For instance see Phys. Rev. A, 87, 012320 (2013)

This should be at least discussed in the text. Better if the authors provide some numerical analysis but this is not mandatory. For sure the reference above need to be included.

In the introduction the authors say:

"However, the performance of both KGPs is restricted by the fundamental rate-loss limit (referred to as PLOB bound) [24,25], which leads to that the key generation rate can only vary with the channel transmittance linearly"

It is imperative to remove [24] Takeoka et al from here. Ref. [24] Takeoka et al is *not* the PLOB bound. Ref. [24] Takeoka et al is instead an old and *incorrect* bound (called TGW bound) that has no operational meaning in these QKD comparisons.

The TGW bound is a substantial wrong result obtained with a fundamental error in its derivation (the TGW authors completely missed to account for technical issues related to the shield system to be considered in the definition of a private state; for more details about the shield system in private states see: IEEE Trans. Inf. Theory 55, 1898 (2009)). This error led them to write a bound which is too large and does *not* represent the actual scaling for QKD.

In the place of [24] the authors must cite the first result on the QKD-capacity scaling, which is:

[2009PRL] Phys. Rev. Lett. 102, 050503 (2009). 

This is the first paper that showed that -log2(1-T) is an achievable rate for QKD (comes from eq. (2) there); then PLOB [25] showed that -log2(1-T) cannot be surpassed, i.e., it is also an upper bound.

This means that [2009PRL] and PLOB [25] together show that -log2(1-T) is exactly the secret-key agreement capacity of a lossy communication channel.

TGW bound [24] is therefore just a useless reference to delete. 

This is the wording I strongly suggest:

"However, the performance of both KGPs is restricted by the fundamental rate-loss limit (referred to as PLOB bound) [2009PRL , 25], which is equal to -log_2(1-T) where T is the channel transmittance; this implies that the key generation rate can only vary with the channel transmittance linearly, asymptotically as 1.44T bits per channel use."

MDI-QKD was co-proposed in Braunstein et al. Phys. Rev. Lett. 2012, 108, 130502 which is the letter before Lo et al. Phys. Rev. Lett. 2012, 108, 130503. Add this reference.

Once the authors have addressed the issues above I can provide a final recommendation.

Author Response

Point 1: I have read this interesting manuscript which contains publishable contents. The idea put forward of a TF-based QDS scheme can have experimental implications. The authors study the protocol assuming the use of two-intensity decoy states. I would like to see the potential impact of using more decoys in terms of performance. Can we have an observable improvement by using a 3-intensity decoy state method? For instance see Phys. Rev. A, 87, 012320 (2013). This should be at least discussed in the text. Better if the authors provide some numerical analysis but this is not mandatory. For sure the reference above need to be included.

Response 1:  According to the comment of reviewer, we cited Wang’s three-intensity decoy state method [Phys. Rev. A, 87, 012320 (2013)] in ref [51] and further performed a fair comparison in Figure 4 between our TF-QDS protocol and the protocols using more decoy states, with the same experimental parameters. Some statements about the simulation results “We perform a numerical simulation of M=8 to evaluate the potential impact of using more decoy states in terms of performance. Figure 4 shows the comparison of simulation results for TF-QDS with different numbers of decoy intensities. The results show that the signature rate and transmission distance of TF-QDS with two decoy states is close to that of three[51] (and four) decoy states. Therefore, for our TF-QDS protocol, the two-intensity decoy state is sufficient for practical usage, and there is no need to introduce more decoy states for longer transmission distances.”were added on lines 219-225.

Point 2: In the introduction the authors say:

"However, the performance of both KGPs is restricted by the fundamental rate-loss limit (referred to as PLOB bound) [24,25], which leads to that the key generation rate can only vary with the channel transmittance linearly"

It is imperative to remove [24] Takeoka et al from here. Ref. [24] Takeoka et al is *not* the PLOB bound. Ref. [24] Takeoka et al is instead an old and *incorrect* bound (called TGW bound) that has no operational meaning in these QKD comparisons.

The TGW bound is a substantial wrong result obtained with a fundamental error in its derivation (the TGW authors completely missed to account for technical issues related to the shield system to be considered in the definition of a private state; for more details about the shield system in private states see: IEEE Trans. Inf. Theory 55, 1898 (2009)). This error led them to write a bound which is too large and does *not* represent the actual scaling for QKD.

In the place of [24] the authors must cite the first result on the QKD-capacity scaling, which is:

[2009PRL] Phys. Rev. Lett. 102, 050503 (2009).

This is the first paper that showed that -log2(1-T) is an achievable rate for QKD (comes from eq. (2) there); then PLOB [25] showed that -log2(1-T) cannot be surpassed, i.e., it is also an upper bound.

This means that [2009PRL] and PLOB [25] together show that -log2(1-T) is exactly the secret-key agreement capacity of a lossy communication channel.

TGW bound [24] is therefore just a useless reference to delete.

This is the wording I strongly suggest:

"However, the performance of both KGPs is restricted by the fundamental rate-loss limit (referred to as PLOB bound) [2009PRL , 25], which is equal to -log_2(1-T) where T is the channel transmittance; this implies that the key generation rate can only vary with the channel transmittance linearly, asymptotically as 1.44T bits per channel use."

Response 2: According to the suggestion of the reviewer, we had removed ref. [24] Takeoka et al from our manuscript and replaced it with [2009PRL] Phys. Rev. Lett. 102, 050503 (2009). Note that its number had been updated to [25] due to the insertion of a new reference in the previous paragraph. Besides, we adopted the description "However, the performance of both KGPs is restricted by the fundamental rate-loss limit (referred to as PLOB bound) [2009PRL, 25], which is equal to -log2(1-T) where T is the channel transmittance; this implies that the key generation rate can only vary with the channel transmittance linearly, asymptotically as 1.44T bits per channel use." suggested by reviewer to elaborate the PLOB bound. These statements were added on lines 42-49.

Point 3: MDI-QKD was co-proposed in Braunstein et al. Phys. Rev. Lett. 2012, 108, 130502 which is the letter before Lo et al. Phys. Rev. Lett. 2012, 108, 130503. Add this reference.

Response 3: Thanks for the reference recommendation of the reviewer. As Braunstein et al's milestone work, we cited this paper [Phys. Rev. Lett. 2012, 108, 130502] in reference [21].

Reviewer 3 Report

The twin-field approach to quantum digitial signature (QDS) is an interesting problem to study, and the authors have done this. It provides a possible practical future realization of QDS, as the technological elements required for it are well within current reach. The technical part appears to be mathematically sound. As such, I endorse the work for publication.

I suggest the authors may consider two points, which should help to better serve the readership: (1) More elaboration can be provided on why the present scheme does not suffer frm loss due to sifting , vis-a-vis QDS with DPRS; (2) I think it is worth stressing that TF-QDS is a kind of MDI-QDS performed at the single-photon level.

Author Response

The twin-field approach to quantum digital signature (QDS) is an interesting problem to study, and the authors have done this. It provides a possible practical future realization of QDS, as the technological elements required for it are well within current reach. The technical part appears to be mathematically sound. As such, I endorse the work for publication.

I suggest the authors may consider two points, which should help to better serve the readership:

Point 1: More elaboration can be provided on why the present scheme does not suffer from loss due to sifting, vis-a-vis QDS with DPRS.

Response 1: Considering the reviewer’s suggestion, we had added some descriptions to explain why the present protocol does not suffer from loss. These statements “The reason is that in our protocol, only two phases are encoded in the code mode and the key generation rate is not restricted by M. On the other hand, the increase of M tightens the upper bound of Eve's side information estimated in the test mode. Therefore, for our protocol, the signature rate increases with M. However, in TF-QDS with DPRS, M phases are encoded in the code mode, so its signature rate tends to zero with the increase of M, which is caused by the filtering factor.” had been added on lines 83-88.

Point 2: I think it is worth stressing that TF-QDS is a kind of MDI-QDS performed at the single-photon level.

Response 2: We stressed that TF-QDS is a kind of MDI-QDS performed at the single-photon level on lines 64-65 as suggested by the reviewer.

Round 2

Reviewer 2 Report

Happy to recommend the publocation of this paper. Authors have addressed my previous concerns.